# Development of a Capsule Robot for Exploring the Colon

**DOI:** 10.3390/mi10070456

**Published:** 2019-07-06

**Authors:** Jinyang Gao, Zenglei Zhang, Guozheng Yan

**Affiliations:** 1Science and Technology on Electronic Test and Measurement Laboratory, North University of China, Taiyuan 030051, China; 2Shanxi Key Laboratory of Advanced Manufacturing Technology, North University of China, Taiyuan 030051, China; 3Department of Instrument Science and Engineering, Shanghai Jiaotong University, Shanghai 200240, China

**Keywords:** Inchworm-like capsule robot, periodic stroke, body length, wirelessly controlled and powered

## Abstract

A tether-less inchworm-like capsule robot (ILCR) is promising to enable a non-invasive exploration of the colon, while existing ILCRs show barely satisfactory movement performance because the colon environment is nonstructural. In this current study, we develop an enhanced ILCR based on a design rule of maximizing the achievable periodic stroke and minimizing the body length, with the aim of improving movement performance. By designing an axial compact expanding mechanism (EM), employing a novel linear mechanism (LM), and integrating a hollow-cylinder-like power source based on wireless power transmission (WPT), the enhanced ILCR achieves a periodic stroke of 38 mm within a small body length of 33 mm. Our experiments show that the EM and LM can work reliably in an ex-vivo colon with a lot of intestinal mucus, and the power source can safely supply a stable working voltage of 3.3 V even in the worst case. Being wirelessly controlled and powered, the enhanced ILCR shows satisfactory movement performance, with velocities of 15.8 cm/min, 12.1 cm/min, and 7.4 cm/min in a transparent tube, a tiled colon, and a suspended colon, respectively, promising to accomplish an exploration for the 1.5-m long colon within 30 min.

## 1. Introduction

A capsule robot (CR), that can be introduced from the mouth/anus and can move actively in the gastrointestinal (GI) tract, is promising to enable a non-invasive diagnosis and treatment for GI diseases [1]. The active movement is normally implemented in two ways, one exploits a magnetic interaction between a permanent magnet onboard the CR and an external controllable magnetic field [2,3,4,5,6], and the other employs a couple of mechanisms driven by actuators [7,8,9,10,11,12]. The magnetic-type CR can be further classified into two subtypes of magnetic-traction and magnetic-rotation, the former features a 3-D movement and has been successfully used for inspecting the stomach [2,3], while the latter aims at exploring a colon based on multiple wedge effects and has shown satisfactory movement velocity in a range of 14.4 to 72 cm/min [4,5,6]. However, the traction force generated by the magnetic interaction is normally less than 300 mN, which is not sufficient to overcome a friction between the CR and the collapsed stomach/colon lumen. Therefore, the stomach/colon must be filled with liquid to reduce the friction when using the magnetic-type CR. As a competitive alternative, the actuators-driven CR features a large traction force and can be directly used to explore the colon that is collapsed in its nature state [7,8,9,10,11,12]. To achieve a high-quality exploration comparable to a traditional colonoscopy, the actuator-driven CR is expected to possess three abilities: an expanding ability for better visualization in folded areas, a two-way movement ability for repeated observation of suspicious lesions, and a parking ability at a designated location for executing functions such as biopsy [13]. An inchworm-like CR (ILCR) is the most typical representative of the actuators-driven type and is equipped with the three abilities by employing two expanding mechanisms (EMs) for expanding and parking, and a linear mechanism (LM) for two-way movement.

Existing ILCRs [10,14,15,16,17,18] employ four different driving ways, as shown in Table 1. The pneumatic-driven ILCR [14,15] uses an air tube to connect an external controllable air source, and it shows satisfactory velocity by designing a large periodic stroke. To improve the adaptability to the colonic bends, its body is made of soft rubber material to obtain sufficient flexibility. However, a friction between the colon lumen and the air tube serves as a resistance to its movement, which limits its travel distance in the colon. The SMA (shape memory alloy)-driven [16] and micropump-driven [17] ILCRs both show poor movement performance because of the slow response speeds of the SMA and micropump. The micromotor-driven ILCR [10,18], which can be wirelessly powered and controlled, is an ideal device for exploring the colon. However, the existing version focuses on how to integrate all the essential modules (including a power source, a telemetry circuit, mechanisms, and tools for diagnosis or treatment) within a small volume but neglects the achievable movement performance. It can be found from Table 1 that the periodic stroke of the existing micromotor-driven ILCR is less than one-half of its body length, thus resulting in poor movement performance.

Maximizing the periodic stroke and minimizing body length has been recognized as an effective design rule for improving the ILCR movement performance [19,20]. In accordance with this rule, this paper develops an enhanced micromotor-driven ILCR, a tether-less device that has a large periodic stroke of 38 mm and a small body length of 33 mm, and can achieve a horizontal velocity of at least 7.4 cm/min. This paper presents the design details of the enhanced ILCR and some experiments for its performance validation. This paper is organized as follows: first, the overall design of the enhanced ILCR is presented. Then, design details of the EM and LM, as well as the onboard power source, are elaborated on, which show how the design rule is practiced. At last, experiments are conducted to assess the performance of the enhanced ILCR, including the mechanical reliabilities of the EM and LM, the power capability of the on-board power source, and the movement performance in an ex-vivo colon.

## 2. Design Overview

### 2.1. Design Considerations

Design considerations of the ILCR for exploring the colon include:The colon diameter ranges from 25 mm to 60 mm, therefore the body diameter of the ILCR should be less than 25 mm to reduce movement resistance, and the maximum expanding diameter of the EM should exceed 60 mm to ensure that the expanding and parking abilities can be implemented effectively.The colon is suspended by connecting to an abdominal wall with a soft mesentery and the colon tissue is viscoelastic, which are both adverse to the ILCR movement. To impair this adverseness, the design parameters of the ILCR must be carefully selected, and the design rule of maximizing the periodic stroke and minimizing the body length has been proven to be effective.The colon has right, left, and sigmoid flexures. To pass these flexures, the body length of the ILCR should not exceed 50 mm [11], or if exceeded, the body should be flexible.The soft colon tissue is easily damaged and the contact safety must be ensured when the ILCR explores the colon.The ILCR should be tether-less because a tether can cause friction and may abrade the colon tissue, therefore the ILCR is best to be wirelessly controlled and powered.

### 2.2. Overall Design of the Enhanced Inchworm-Like Capsule Robot (ILCR)

To fulfill the above design considerations, an enhanced ILCR is developed, as shown in Figure 1. It is composed of two identical EMs for expanding the intestine, a LM between the two EMs for generating a periodic stroke, several contact devices for improving the contact safety [21], a hollow-cylinder power source around the LM for wireless power induction, a commonly used camera for inspection [22], and a telemetry circuit for wireless communication [10], with geometric parameters as listed in Table 2. The two EMs adopt a previously verified design of an arc-shaped leg to obtain a large diameter range [10,11], but the mechanism for actuating the leg is redesigned for reliability improvement and axial length reduction. The LM adopts a novel design of using one motor to actuate two pairs of lead-screws and nuts, and this design doubles the periodic stroke without increasing the LM length. The contact device is installed at the tip of the arc-shaped leg, and it increases the contact area with the intestine, thus avoiding local stress concentration and lowering the risk of the intestine being expanded excessively. The hollow-cylinder power source takes full advantage of the space around the LM, thus overcoming a limitation of a solid-cylinder power source which increases the ILCR’s body length [23]. Due to the well-designed EM, LM, and power source, the enhanced ILCR achieves a large periodic stroke of 38 mm within a small body length of 33 mm (camera and telemetry circuit not included). The following presents the design details of the enhanced ILCR.

## 3. Design of the Enhanced ILCR

### 3.1. Design of the Expanding Mechanism (EM)

Figure 2a shows the overall design of the EM, it is mainly composed of a planetary reducer motor (PRM) that is waterproof, a baffle I, a baffle II, an annular gear (AG) I, an AG II, three pairs of arc-shaped legs, and some gears for torque transmission, with design parameters as listed in Table 3. Figure 2b shows the working principle of the EM. The output shaft of the PRM is connected to gear I, thus the output torque of the PRM can be transmitted to AG I only by gear I, and can be transmitted to AG II by gears I, III, IV, and II in sequence. The outer surface of AG I has three vaulted bulges that are used for hinging legs a, c, and e. Furthermore, AG II has three identical bulges for hinging legs b, d, and f. When gear I rotates in the direction of the red curved arrow, the EM closes; otherwise, the EM expands. Benefitting from the novel design of hinging the legs at the outer surfaces of AG I and AG II, the axial length of the EM has been reduced by 1.5 mm, as compared to a previously developed EM in [11], where the legs are hinged at the lateral surface of the AG. In addition, because AG I and AG II are rotated at the same speed and in the opposite direction, the legs could expand along a radial locus, which is much safer than another EM used in a previously developed inchworm-like robot [10], where the legs expand along an oblique locus and have a risk of scratching the intestine. Figure 2c shows the installation of the contact device, it is fixed at a triangular part that is set between two legs. This installation has been proven to be reliable, and to have no interference to the leg’s action. To ensure that the EM has a high reliability, solid intestinal contents, which may cause a mechanical jam, are not allowed to access the gears. Figure 2d depicts a solution to this problem. By setting balls with 0.5-mm diameter between each two of baffle I, AG I, AG II, and baffle II, gaps between them are set to be 0.1 mm, a figure that is small enough to prevent all the solid intestinal contents. In addition, the balls reduce friction and mechanical wear when AG I and AG II rotate, thus ensuring high mechanical efficiency and long-term reliable operation. Note that the intestinal mucus can still get into the EM, but because the EM is a low-speed mechanism, the caused viscous resistance is negligibly small and has little effect on the EM’s operation. Figure 2e shows the fabricated prototype of the EM, whose mechanical parts were made of stainless steel 304, using an Electrical Discharge Machining (EDM) process. The expanding force of the EM prototype has been measured using the method in [11], which exceeds 3 N in the diameter range of 24 mm–56 mm and decreases rapidly from 3 N to 0.8 N in the diameter range of 56 mm–61 mm, manifesting that the developed EM can expand the collapsed colon at most cases.

### 3.2. Design of the Linear Mechanism (LM)

Figure 3 shows the design details of the LM, which is mainly composed of a PRM, two pairs of lead-screws and nuts, and three gears for torque transmission, with design parameters as listed in Table 4. Figure 3a shows its working principle. The PRM’s output shaft is connected to gear V, thus the output torque can be transmitted to lead-screw I by gears V and VI and can be transmitted to lead-screw II by gears V and VII. Because lead-screw I has a right-hand thread and lead-screw II has a left-hand thread, nuts I and II will be linearly moved in the opposite direction when the PRM rotates. The two PRMs for actuating the two EMs are connected to nuts I and II, respectively, thus elongation and retraction of the ILCR can be implemented. Benefiting from this novel design, the LM achieves a large periodic stroke of 38 mm, while only occupying an axial length of 26 mm.

The sealing design is necessary for the LM because once intestinal mucus gets into it, cleaning can be troublesome, and the power capability of the power source can be lowered significantly. Considering the sealing object is the linearly-moved PRM and the sealing-caused mechanical loss should be minimized, an O-ring made of Viton is adopted here, as shown in Figure 3b. The O-ring is set at a L-groove of a mounting plate, and the sealing surfaces are the outer surface of the PRM and the inner surface of the L-groove. To avoid axial movement of the O-ring, a cover plate is set next to the mounting plate. Finally, by fitting an ABS-plastic shell around the two cover plates, sealing for both the LM and power source is implemented. To evaluate the feasibility of the proposed sealing method, axial thrust loss of the LM caused by sealing is calculated:(1)$$f=f_{O}+f_{P}$$
where f is the total thrust loss; $f_{O}$ is the thrust loss caused by the friction between the O-ring and the PRM; and $f_{P}$ is the thrust loss caused by the negative pressure which is formed when the LM elongates. Referring to Figure 4 which shows the contact state between the O-ring, PRM, and L-groove, $f_{O}$ can be calculated as:(2)$$f_{O}=2\pi \mu d_{i}\int_{0}^{b}p(x)dx$$
where μ is the friction coefficient between the O-ring and the PRM, and it is measured as 0.12; $d_{i}$ (= 6.3 mm) is the inner diameter of the O-ring and it is equal to the diameter of the PRM; b is the contact width; and p(x) is the contact pressure distribution along the contact width. Referring to the Lindley formula and Hertz contact theory, calculation formulas of b and p(x) can be deduced out:(3a)$$b=d\sqrt{\frac{7.5C^{1.5}+300C^{6}}{\pi }}$$
(3b)$$p(x)=2p_{0}\sqrt{\frac{x}{b}(1-\frac{x}{b})}$$
(3c)$$p_{0}=\frac{10^{6}d(5C^{1.5}+200C^{6})(15.75+2.15H_{a})}{\pi b(100-H_{a})}$$
where C is the compression ratio; d is the wire diameter of the O-ring; $p_{0}$ is the contact pressure peak; and $H_{a}$ is the hardness of the O-ring, and it takes a large value of $H_{a}$ = 80 to lower the risk of sealing failure caused by pressure pulsation. With an aim of minimizing $f_{O}$, C and d have been optimized with Equations (2) and (3), and the optimized results are: C = 5%, d = 1 mm. Correspondingly, the contact pressure peak $p_{0}$ = 1023 kPa, the contact width b = 0.1634 mm, and the friction-caused thrust loss $f_{O}$ = 0.6232 N. Note that $p_{0}$ is much larger than the medium pressure (≤200 kPa [24]) in the colon, thus will ensure a reliable sealing.

The thrust loss $f_{P}$ caused by the negative-pressure can be calculated as:(4a)$$f_{P}=S_{m}(p_{0}^{air}-p_{1}^{air})$$
(4b)$$p_{1}^{air}=V_{0}^{air}p_{0}^{air}/(V_{0}^{air}+V_{\Delta })$$
where $S_{m}$ (= 31.17 mm^2^) is the cross-section area of the PRM, $p_{0}^{air}$ (= 10^5^ Pa) is the standard atmospheric pressure, $p_{1}^{air}$ is the internal pressure of the LM when the two PRMs fully elongate, $V_{0}^{air}$ (= 1912.8 mm^2^) is the volume of free space within the LM when the two PRMs fully retracts, and $V_{\Delta }$ (= 1184.6 mm^2^) is the volume of the two PRMs. From Equation (4), $f_{P}$ can be calculated as $f_{P}$ = 1.1922 N.

Figure 3c shows the fabricated prototype of the LM, its sealing performance has been verified as good with a water-proof test, and the test method is detailed in [10]. Its thrust is measured as 4.57 N, even if the sealing caused a total thrust loss of 1.18 N. Therefore, the sealing design for the LM is reliable and feasible.

### 3.3. Design of the Power Source

The onboard power source is designed to be a hollow-cylinder to take full advantage of the free space around the LM. Considering that existing transmitting coil can only excite a unidirectional alternating magnetic field [25], the power source should consist of three mutual-orthogonal windings to ensure that it can induce electric power in any orientation. Figure 5 shows the employed power source in this study, it is hollow-cylinder-like composed of a ferrite core, and three windings a, b, and c, with design parameters as listed in Table 5. The ferrite core has four lateral grooves on its outer surface, and each two opposing grooves are used for setting winding a or b. Windings a and b adopt identical design, and they both have two saddle-shaped sub-windings. Note that the winding direction of the two sub-windings should be the same to ensure that the induced electromotive forces (EMFs) can be superposed correctly. Winding c is wound around the outer surface of the ferrite core. The two cover plates on both sides of the power source can cause an eddy-current effect, which will increase the equivalent series resistance (ESR) of winding c. Therefore, winding c is set at the center of the ferrite core, where the ESR increment has been verified to be minimal.

When placing the power source in an alternating magnetic field, windings a, b, and c will induce EMFs with maximums of $\varepsilon_{a}$, $\varepsilon_{b}$, and $\varepsilon_{c}$, respectively:(5)$$\varepsilon_{i}=\sqrt{2}\pi fI_{m}M_{i}\;i=a,b,c$$
where f and $I_{m}$ are the frequency and amplitude of the current flowing in the transmitting coil, respectively. This study uses a transmitting coil of double-layer-solenoid type [26], for which f = 218 kHz and $I_{m}$ = 1.98 A. $M_{i}$ is the mutual inductance when the equivalent plane of winding i is perpendicular to the alternating magnetic field direction. For consideration of the symmetry of the power source, $\varepsilon_{a}=\varepsilon_{b}=\varepsilon_{c}$ is desired, which requires:(6)$$M_{a}=M_{b}=M_{c}\Rightarrow N_{a}M_{a0}=N_{b}M_{b0}=N_{c}M_{c0}$$
where $N_{i}$ and $M_{i0}$ are the number of turns and single-turn mutual inductance of winding i (i=a,b,c), respectively. $M_{i0}$ is certain when the ferrite core size is given, and it can be obtained with a common method of measuring the open-circuit EMF of winding i. When $\varepsilon_{a}=\varepsilon_{b}=\varepsilon_{c}$ holds and windings a, b, and c are connected in parallel, the power source in any orientation can induce an EMF in a range from $\sqrt{3}\varepsilon_{i}/3$ to $\varepsilon_{i}$. Because the power source is required to supply a stable working voltage for the ILCR even in the worst case, it can be equivalent to a voltage source with an EMF $\varepsilon_{\min}$ (=$\sqrt{3}\varepsilon_{i}/3$) in series with a maximal ESR $R_{\max}$ (= $\max[R_{a},R_{b},R_{c}]$), as shown in Figure 6. By analyzing the on-off character and power loss of the full-bridge rectifier, the average of the input voltage of the LDO regulator can be derived as:(7)$$V_{I}=\frac{2\sqrt{2}\pi \varepsilon_{\min}R_{LDO}-16V_{FD}R_{LDO}}{8R_{LDO}+\pi^{2}R_{\max}}$$
where $R_{LDO}$ is the input resistance of the LDO regulator, $V_{FD}$ (= 0.27 V) is the forward voltage of the diode. $R_{LDO}$ can be expressed as:(8)$$R_{LDO}=\frac{R_{L}}{\eta_{LDO}}\;\overset{\eta_{LDO}=V_{O}/V_{I}}{\to }\;R_{LDO}=\frac{R_{L}V_{I}}{V_{O}}$$
where $R_{L}$ is the ILCR load, $\eta_{LDO}$ is the efficiency of the LDO regulator, and $V_{O}$ is the working voltage of the ILCR. To avoid excessive power dissipated by the LDO regulator, we assume $\eta_{LDO}$
$=V_{O}/V_{I}$
≥ 50%. To obtain the required $V_{O}$, $V_{I}\geq V_{O}+0.3\;V$ should hold because the voltage drop of the LDO regulator is normally 0.3 V. Thus, forming a constraint of $V_{O}+0.3\;V$
$\leq V_{I}$
$\leq 2V_{O}$. By substituting Equations (7) and (8) into this constrain, it has:(9)$$V_{O}+0.3\;V\leq \frac{\sqrt{2}}{4}\pi \varepsilon_{\min}-2V_{FD}-\frac{\pi^{2}R_{\max}V_{O}}{8R_{L}}\leq 2V_{O}$$

In Equation (9), $V_{O}$ (normally 3.3 V), $V_{FD}$ (= 0.27 V), and $R_{L}$ (≈15 Ω, the maximal power demand of the ILCR is about 730 mW @ 3.3 V) are given, $\varepsilon_{\min}$ and $R_{\max}$ are closely related to the number of turns $N_{i}$ and wire diameters $d_{i}$ of winding i (i=a,b,c). Therefore, Equation (9) can be used as a guidance for parameters selection. Complying with Equation (9), $N_{i}$ and $d_{i}$ have been selected, as listed in Table 5. Using these parameters, a power source prototype shown in Figure 5b is fabricated, which can supply a $V_{I}$ = 6.38 V in the worst case, ensuring the ILCR can work at a stable voltage of 3.3 V all the time.

## 4. Experiments

The performance of the enhanced ILCR was assessed using the experimental setups shown in Figure 7. In the experiment, the ILCR was wirelessly powered by a transmitting coil on a driving box and was wirelessly controlled by a HMI and a command/data transceiver. The transmitting coil employs a double-layer-solenoid structure, which makes a good compromise between the intensity and uniformity of the alternating magnetic field, as compared to the solenoid, segmented solenoid, and Helmholtz types [27]. The driving box contains a full-bridge inverter for DC-AC conversion, a square-wave generator for controlling the inverting frequency, and a vacuum capacitance for making the transmitting coil resonate. It is responsible for supplying a stable AC voltage for the transmitting coil and features a strong load capacity (15 V, 3 A). The command/data transceiver is connected to the HMI with a serial port line, and its wireless link with the telemetry circuit onboard the ILCR is implemented by employing a wireless communication chip Si4455. Being wirelessly powered and controlled, the ILCR’s performance was tested in a transparent tube, a suspended colon, and a tiled colon, in sequence. Note that the three tests were all conducted in the horizontal direction, this is due to the fact that although the in-vivo colon has an angle of slope ranging from 0° to 90°, it can be adjusted to be almost horizontal by controlling the patient postures.

The test in the transparent tube was mainly to assess the cooperative working performance of each module in the ILCR. The ILCR was observed to respond to the command issued from the HMI within 1 s and simulate inchworm movement smoothly, as referred to the Appendix A. Figure 8 shows the movement snapshots in two cycles, during which the illumination LED of the camera (required working voltage ≥ 3 V) shone stably. Each cycle contains four gaits and the gait transition was implemented with a closed-loop control scheme that employed the running current of the PRM as a feedback [17]. In each cycle that lasts about 14 s, the ILCR advanced 36.8 mm, thus its velocity was calculated as 15.8 cm/min. These results show the telemetry circuit, EM, LM, camera, and power source, cooperate well after being integrated. Considering a colonoscopy procedure lasts about 30 min and can be longer if a surgical operation is included, 1-h reliability of the ILCR is tested. During the 1-h test, the ILCR was controlled to work continuously, and the control command was reissued each 2 min. The ILCR responded correctly 29 times to the issued 30 commands. The EM and LM had no mechanical failure except the LM failed to fully elongate 3 times, which may be caused by the installing and machining errors of the mechanical parts. The power source supplied stable working voltage for the ILCR all the time and its temperature was measured with a short-wave infrared thermometer (AR872D+) every minute. Figure 9 shows the measured temperature curve, which indicated that the temperature rose rapidly in the first 8 min, then trended to a stable value of 38.8 °C that is lower than the safety limit of 42.5 °C [28]. To further validate the safety of the power source, its temperature curve when continuously operating for 1 h in an ex-vivo colon was also measured and plotted in Figure 9, which indicated that the temperature rose rapidly in the first 11 min, then trended to a stable value of 36.9 °C. The difference between the two temperature curves may be related to that the colon tissue has a higher heat absorbance rate than the transparent tube which is made of acrylic plastic. These results validate that the power source will not cause a heat injury to the colon’s tissue. After the 1-h test, the sealing performance of the LM was verified still good with a second water-proof test. The abrasion of the EM and LM was negligibly small except the arc-shaped legs swung slightly in the axial direction.

The tests in the suspended and tiled colons were used mainly to assess the movement performance of the enhanced ILCR. These two test environments represent two extremes of the in-vivo colon, which is freely suspended by connecting to the mesentery but is also constrained by the compression from the abdominal fat. Here, the suspended colon was fully free and the tiled colon was fully constrained. The velocities in the suspended and tiled colon were respectively measured as 7.4 cm/min and 12.1 cm/min, which were both obviously lower than the 15.8-cm/min velocity in the transparent tube. This was much related to the viscoelastic and slippery properties of the colon. The viscoelastic colon was observed to be stretched/compressed when the ILCR elongated/retracted. The slippery colon lumen lowered the traction force and the expanded EM was observed to slip slightly. These both resulted in stroke losses and therefore lowered the velocity. Note that the stretched/compressed deformation of the suspended colon was much more significant compared to that of the tiled one, and this accounts for the velocity difference in this two test environments. However, the velocity of 7.4 cm/min was still obviously higher than the velocities of other existing tether-less ILCRs listed in Table 1. After these two tests, the sealing performance of the LM was verified as still good with a third water-proof test. Then, the EM and LM were disassembled: a mass of intestinal mucus leaked into the EM but did not cause a mechanical jam, trace amounts of intestinal mucus were found on both sides of the O-ring but not found inside the LM, further confirming the sealing design was feasible and reliable.

## 5. Conclusions

In this paper, a tether-less enhanced ILCR for exploring the colon, is presented. The design rule of maximizing periodic stroke and minimizing body length has been nicely fulfilled by the well-designed EM, LM, and power source, and the enhanced ILCR achieves a large periodic stroke of 38 mm within a small body length of 33 mm. The two methods of setting the gaps between mechanical parts of the EM to 0.1 mm and sealing the LM with two O-rings, have been verified feasible to ensure the EM and LM work reliably in the colon that has high levels of intestinal mucus. The developed power source with design parameters that satisfy equation (9), can supply a stable voltage of 3.3 V to the ILCR even in the worst case, and does not cause a heat injury to the colon’s tissue. Our experiments show that the enhanced ILCR works properly and reliably when being wirelessly powered and controlled, and its movement performance was better than other existing counterparts, with a satisfactory velocity of 7.4 cm/min in a freely suspended colon. Future work will focus on analyzing the influence of the colon physiologic motion (e.g., segmentation, peristalsis, and conditioned reflex stimulated by the ILCR) on the ILCR’s movement performance, and the analysis results will be used for further optimization of the ILCR’s design parameters (e.g., the expanding force of the EM, the axial thrust of the LM, and the periodic stroke).

## Figures and Tables

**Figure 1 micromachines-10-00456-f001:**
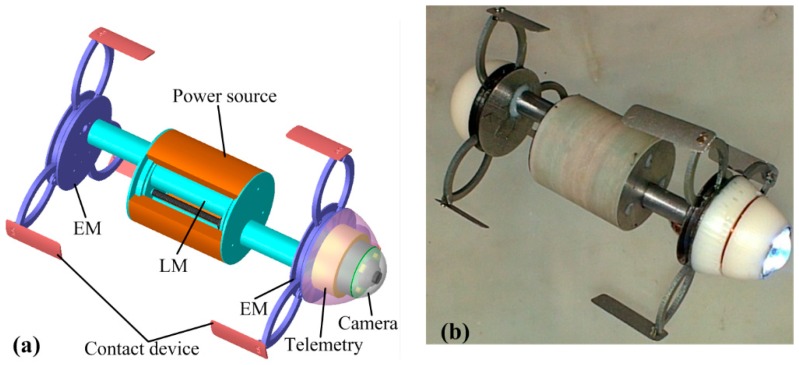
Design showing of the enhanced inchworm-like capsule robot (ILCR): (**a**) the CAD design; (**b**) the fabricated prototype.

**Figure 2 micromachines-10-00456-f002:**
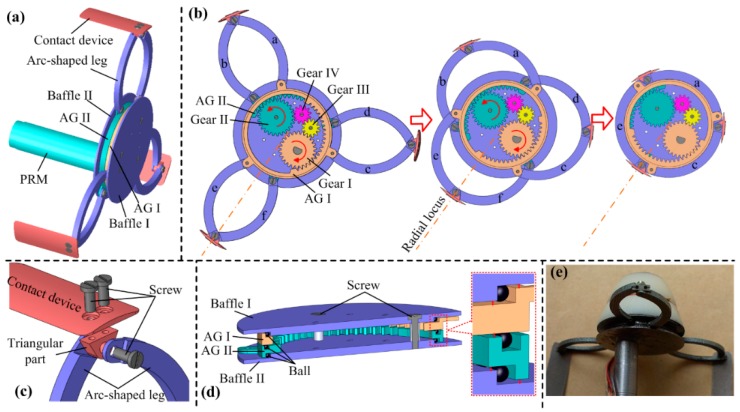
(**a**) Overall design of the expanding mechanism (EM); (**b**) working principle of the EM; (**c**) installation of the contact device; (**d**) cutaway view of EM, which shows the installations of AG I and AG II; (**e**) fabricated prototype of the EM.

**Figure 3 micromachines-10-00456-f003:**
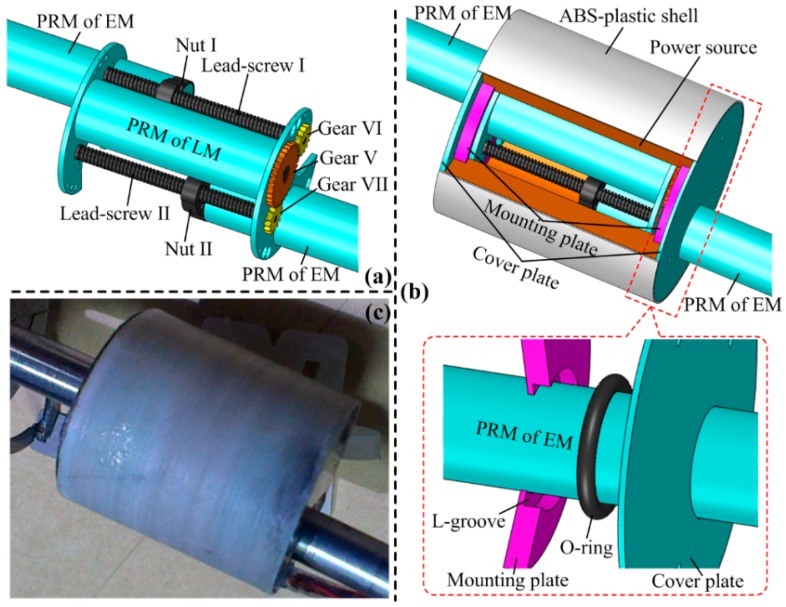
(**a**) Working principle of the linear mechanism (LM); (**b**) sealing design of the LM; (**c**) fabricated prototype of the LM.

**Figure 4 micromachines-10-00456-f004:**
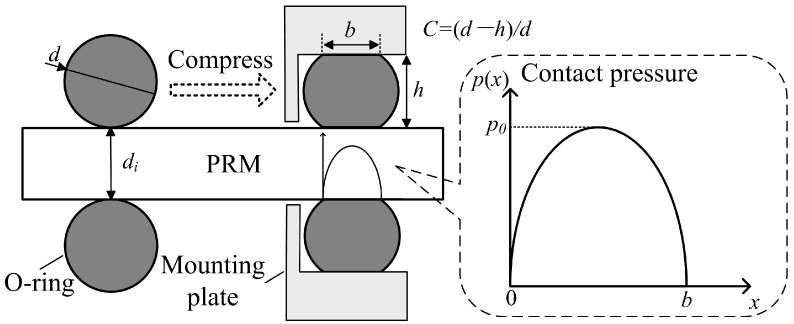
Contact state between the O-ring, planetary reducer motor (PRM), and L-groove.

**Figure 5 micromachines-10-00456-f005:**
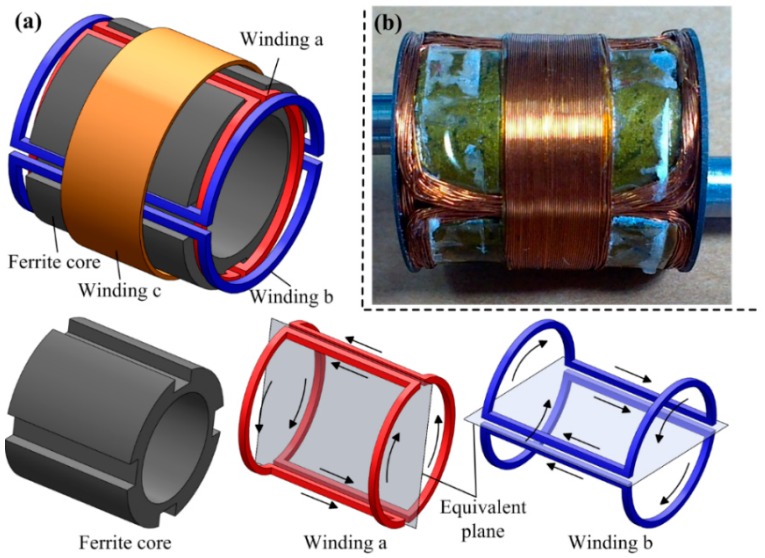
(**a**) Design showing of the hollow-cylinder-like power source, which is composed of a ferrite core, two saddle-shaped windings a and b (arrows indicate winding direction), and a circular winding c; (**b**) a fabricated prototype of the power source for integration.

**Figure 6 micromachines-10-00456-f006:**
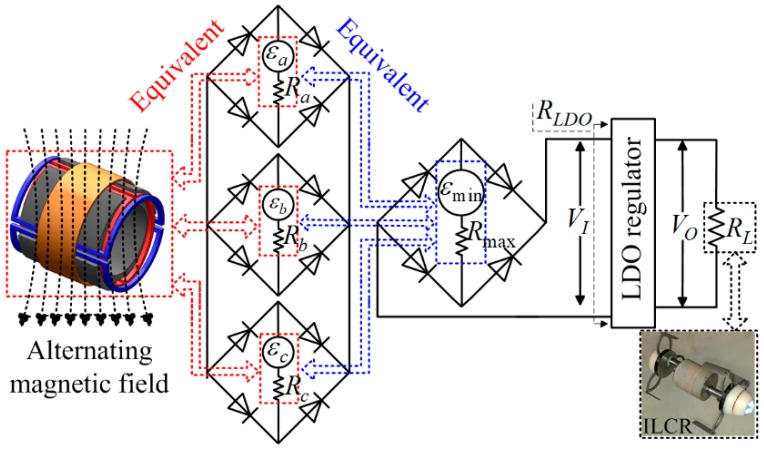
Circuits for converting the induced EMFs of windings a, b, and c to a steady DC voltage for powering the ILCR. The induced EMFs $\varepsilon_{a}$, $\varepsilon_{b}$, and $\varepsilon_{c}$ are firstly rectified by three full-bridge rectifiers that each is composed of four Schottky diodes, then superimposed in parallel, and finally regulated by a LDO regulator. The red and blue dotted lines indicate the equivalent process of the power source.

**Figure 7 micromachines-10-00456-f007:**
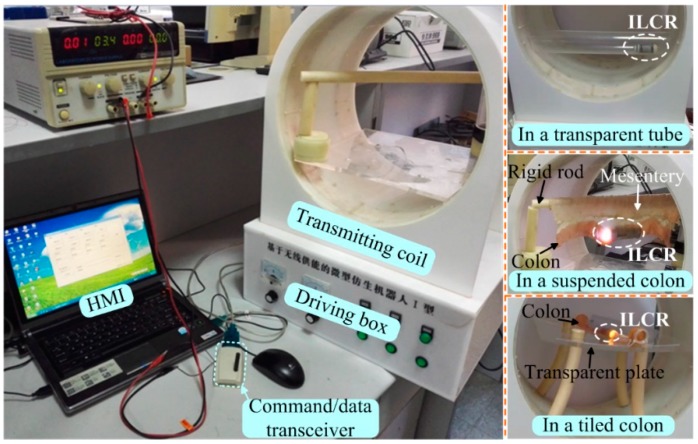
Experimental setup for testing the performance of the enhanced ILCR, and the test environments include a transparent tube, a suspended colon that is connected to a soft mesentery, and a tiled colon that is placed on a transparent plate.

**Figure 8 micromachines-10-00456-f008:**
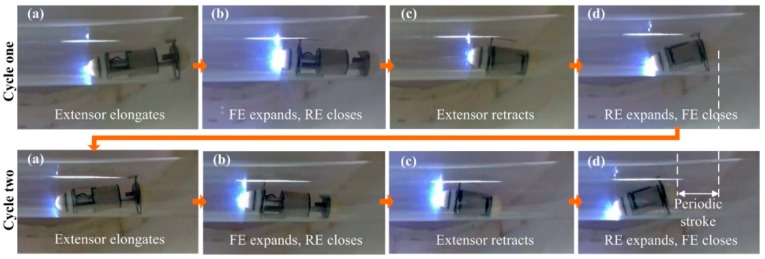
Movement snapshots of the enhanced ILCR in two cycles. Each cycle contains four gaits of (**a**–**d**) that last about 6 s, 1 s, 6 s, and 1 s, respectively. In the gait sequence of (**a**–**d**), the ILCR moves forward; in the reverse sequence, the ILCR moves backward.

**Figure 9 micromachines-10-00456-f009:**
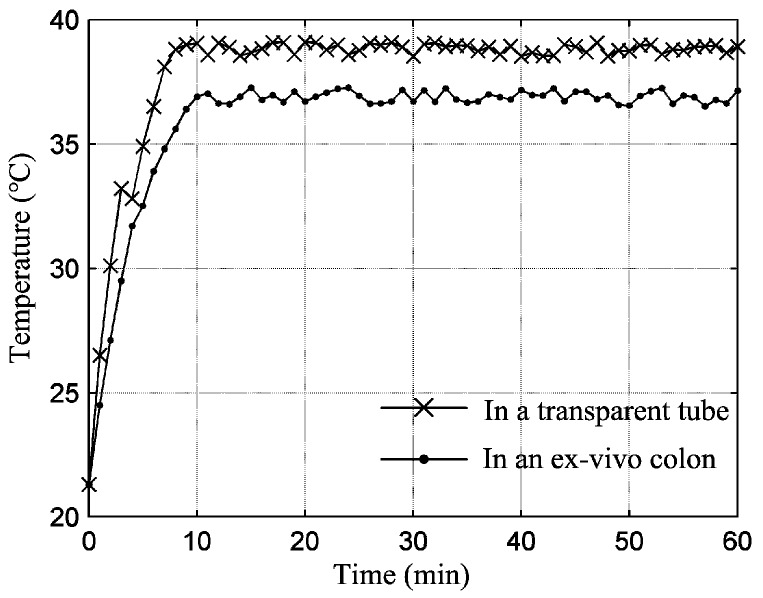
Changing of the temperature of the power source when it supplied electric power to the enhanced ILCR continuously.

**Table 1 micromachines-10-00456-t001:** Main features of inchworm-like capsule robots (ILCRs).

Driving Ways	Ref.	Tether	Power Supply	Communication and Control	Periodic Stroke (mm)	Body Length (mm)	Diameter (mm)	Horizontal Velocity (cm/min)
Pneumatic-driven	[14]	Yes	Air tube		115	115	24	9
	[15]	Yes			140	110	15	51
SMA-driven	[16]	Yes	Power line		-	40	15	3
Micropump-driven	[17]	No	WPT	Wireless	16.5	73	20	1.6
Micromotor-driven	[18]	No	WPT	Wireless	44	128	17	3
	[10]	No	WPT	Wireless	10.5	27	14	4.2
	This work	No	WPT	Wireless	38	33	24	≥7.4

**Table 2 micromachines-10-00456-t002:** Geometric parameters of the enhanced ILCR.

Overall size	Diameter	24 mm
	Body length	33 mm
Two EMs	Diameter range	24–61 mm
	Length	3.5 mm
LM	Diameter	16 mm
	Length	26 mm
	Periodic stroke	38 mm
Contact device	Length × Width × Thickness	15 mm × 7 mm × 0.5 mm
Power source	Diameter range	16–23.5 mm
	Length	25 mm
Camera [22]	Diameter	12 mm
	Length	8 mm
Telemetry circuit [10] (housed in a copper shell)	Diameter	14 mm
	Length	4 mm

**Table 3 micromachines-10-00456-t003:** Design parameters of the expanding mechanism (EM).

PRM (waterproof)	Size	Φ 6.3 mm × 24.5 mm
	Output torque	452 gf·cm
	Working voltage	3.3 V
	Rated/Stall current	40/190 mA
AG I/II	Teeth number (incomplete)	55 (20)
	Modulus	0.2 mm
Gear I/II/III/IV	Teeth number	26/26/12/12
	Modulus	0.2 mm
Arc-shaped leg	Arc-diameter	22 mm
	Radian	0.594π

**Table 4 micromachines-10-00456-t004:** Design parameters of the Linear Mechanism (LM).

PRM	Size	Φ 6.3 mm × 19 mm
	Output torque	280 gf·cm
	Working voltage	3.3 V
	Rated/Stall current	40/190 mA
Gear V/VI/VII	Teeth number	26/26/12/12
	Modulus	0.2 mm
Lead-screw I/II	Thread direction	Right-hand/left-hand
	Nominal diameter	2 mm
	Thread angle	60°
	Thread pitch	0.4 mm
Nut I/II	Axial length	2.5 mm

**Table 5 micromachines-10-00456-t005:** Design parameters of the power source.

Ferrite core	Overall size	Φ (16~23) mm × 20 mm
	Lateral groove size	20 mm × 4 mm × 2 mm
	Material/Permeability	Mn-Zn/R6K
Winding a	Single-turn mutual inductance $M_{a0}$	1.37 × $10^{-7}$ H
	Number of turns $N_{a}$	68
	Wire diameter $d_{a}$	0.15 mm
	ESR $R_{a}$	16.34 Ω
Winding b	Single-turn mutual inductance $M_{b0}$	1.54 × $10^{-7}$ H
	Number of turns $N_{b}$	60
	Wire diameter $d_{b}$	0.15 mm
	ESR $R_{b}$	13.56 Ω
Winding c	Single-turn mutual inductance $M_{c0}$	1.82 × $10^{-7}$ H
	Number of turns $N_{c}$	52
	Wire diameter $d_{c}$	0.2 mm
	ESR $R_{c}$	9.73 Ω

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
