# Peer review of "Development of a Capsule Robot for Exploring the Colon"

_micromachines, 2019, doi:10.3390/mi10070456_

Round 1
Reviewer 1 Report
Very good and impressive work. The paper could still be improved by working on the following points:
line 35 The reference should be added in the brackets.
line 39 "satisfactory movement speed" could you be more specific and give a value.
line 73 In table 1, it would be interesting add a line with your design targets. In this way, we can immediately see how your work compare to the state of the art.
line 82 not clear what you mean with "suspended properties".
line 117 it would be better to put table 3. after line 120, where PRM is defined.
line 141 When you say that " the intestinal mucus ... has little effect on the EM", it's not clear at this point if it has been tested or if it is a supposition
line 224 The schematic is not clear. The concept is understandable, but I think that the connection to the LDO is not correctly drawn. It should be connected perpendicular to the coil.
Line 239 « at any attitude » do you mean in any orientation?
Line 262, section experiments:
- For me it’s not clear what a suspended colon and a tiled colon are.
- Could you also give indications about the orientation, in which the ILCR was tested? Moving up, down, horizontal, all of them?
- The goal of this work is to improve the velocity of the ILCR. In your experiments you focused on the step size of the ILCR, which is only one aspect of the velocity. Could you say something about the step frequency?
Line 296 Do you have any clue why the LM failed 3 times to fully elongate?
Author Response
Response to Reviewer 1 Comments
Point 1: Very good and impressive work. The paper could still be improved by working on the following points:
Response 1: Thank you very much for your comments and approval on our work.
Point 2: Line 35, the reference should be added in the brackets.
Response 2: The reference has been added in the brackets.
Point 3: Line 39, "satisfactory movement speed" could you be more specific and give a value.
Response 3: As your suggestion, “satisfactory movement speed” has been more specific. The magnetic-rotation CRs in references [4], [5], and [6] have movement velocities of 6.35 mm/s, 2.4 mm/s, and 12 mm/s, respectively. Therefore, the “satisfactory movement speed” has been specific in a range of 14.4 to 72 cm/min in the revised manuscript.
Point 4: In table 1, it would be interesting add a line with your design targets. In this way, we can immediately see how your work compare to the state of the art.
Response 4: It is a very good suggestion, we have added a line to show the design specifications of the enhanced ILCR in our work in the revised table 1. In addition, a sentence has been added in line 67 to make the text content in consistent with the revised table 1.
Point 5: Line 82, not clear what you mean with "suspended properties".
Response 5: We are very sorry for not clearly explaining the “suspended property”. The colon is connected to the abdominal wall by a mesentery (referring to Figure 7 “In a suspended colon”). Because the mesentery is soft, the colon can be considered to be suspended. The “suspended property” has been proven to cause a stroke loss when the ILCR retracts in Ref [14]. We have modified the sentence in line 83 to make the “suspended property” more clear.
Point 6: Line 117, it would be better to put table 3. after line 120, where PRM is defined.
Response 6: As your suggestion, table 3 has been put after the line where the PRM is defined.
Point 7: Line 141, when you say that " the intestinal mucus ... has little effect on the EM", it's not clear at this point if it has been tested or if it is a supposition.
Response 7: The intestinal mucus is a non-Newtonian fluid. When it gets into the EM (note that only the intestinal mucus can gets into the EM, and all of the other solid intestinal contents can be prevented by the adopted mechanical design, see lines 137 to 140), the caused viscous resistance is negligibly small because the EM is a low-speed mechanism. Therefore we say that “the intestinal mucus… has little effect on the EM operation”. We have revised this sentence to make the statement reasonable (see lines 142 to 144). In fact, the intestinal mucus has little effect on the EM operation has also been verified in our experiment (see lines 336 to 338).
Point 8: Line 224, the schematic is not clear. The concept is understandable, but I think that the connection to the LDO is not correctly drawn. It should be connected perpendicular to the coil.
Response 8: To make the schematic (Figure 6) clear, we have added some explanation in the figure caption. In addition, we are very sorry for our carelessness to make an incorrect connection between the full-bridge rectifier and the LDO regulator, and have corrected it.
Point 9: Line 239, « at any attitude » do you mean in any orientation?
Response 9: The words “at any attitude” do mean in any orientation. And we have replaced “at any attitude” with “in any orientation” in the revised manuscript.
Point 10: Line 262, section experiments:
(1) For me it’s not clear what a suspended colon and a tiled colon are.
(2) Could you also give indications about the orientation, in which the ILCR was tested? Moving up, down, horizontal, all of them?
(3) The goal of this work is to improve the velocity of the ILCR. In your experiments you focused on the step size of the ILCR, which is only one aspect of the velocity. Could you say something about the step frequency?
Response 10:
(1) The suspended colon is connected to a soft mesentery, and the other end of the mesentery is fixed at a rigid rod, as shown in Fig. 7; and this connection makes the colon in a suspended state. The tiled colon is directly placed on a transparent plate, as shown at the lower right of Fig. 7; because a friction exists between the colon and the transparent plate, the tiled colon is more restricted compared to the suspended colon. We have added some necessary indications in Figure 7, and a brief explanation has also been added to the caption of Figure 7 (see lines 287 and 288).
(2) The ILCR was tested only in the horizontal direction in our study. In fact, the velocities of the existing ILCRs listed in Table 1 were all measured in the horizontal direction. This is because although the in-vivo colon may has an angle of slope ranging from 0° to 90°, the angle of slope can be reduced to be less than 5° (almost horizontal) by adjusting the posture of the patients [11]. To clarify the orientation, we have added a word “Horizontal” in the last column of Table 1 and a sentence “Note that … patient postures” in lines 281 to 283.
(3) As your suggestion, we have added some more information about the velocity of the ILCR (see the sentence “In each cycle that lasts about 14 s … was calculated as 15.8 cm/min” in lines 294 and 295) in the revised manuscript. Here, we do not use the word of “frequency” because the duration time of each cycle is larger than 1 s. In addition, the duration time of gaits (a), (b), (c), and (d) is about 6 s, 1 s, 6 s, and 1 s, respectively; and we have also added this information in the caption of Figure 8.
Point 11: Line 296, do you have any clue why the LM failed 3 times to fully elongate?
Response 11: Dear reviewer, to be honest, we also can not explain exactly why the LM failed 3 times to fully elongate. In fact, the failed times may be random, it can be 1, 2, 3, 4…, but during the 1-hour test, the LM happened to fail 3 times. We infer this failure may be related to the employed control scheme for gait transition. In our study, the gait transition to a next gait is implemented by detecting the running current of the PRM, and once the running current exceeds a limit, the ILCR is controlled to execute the next gait. In normal cases, the running current exceeds the limit only at the end of a gait, e.g., at the end of gait (a) when the extensor fully elongates. However, because of the mechanical uncertainty caused by installing and machining errors, a mechanical jam may happen when the extensor has not fully elongated, which can also make the running current exceeds the limit and cause a gait transition. This may explain why the LM failed 3 times to fully elongate. We have added relevant description in the revised manuscript (see lines 293 and 294, see lines 301 and 302).

Reviewer 2 Report
A wireless capsule robot to explore colon of GI Tract is proposed. Powering the system in a wireless manner and achieving high stroke are the two highlights of the proposed system.
To improve the paper further, following comments can be considered:
In two different places, authors mention "invasive exploration". Is it really invasive? Or should it be called "non-invasive"? In gastroloenterology, GI Tract is still outside of the body.
In the intro, there is a reference parenthesis without references (line 35).
In intro, authors mention about magnetic control and then jump to mechanisms driven by actuators. If you are going to choose latter one, it would be good to explain its disadvantages compared to magnetic control case.
In Fig 2, a lot of gears are shown. It would be helpful if dimension details of the smallest component is shown and also how these small gear components are manufactured?
What is the brand and model name of the motors which have been used in this study.
Line 159 -161: Is there any disadvantage of using two direction extension idea?
Videos of experiments should be added as supplementary videos.
Can you test your system for vertical or diagonal colon orientation case to show it will work in real body in the future?
In line 317-320: leaking and jamming are discussed. But was it very similar to real in-vivo environment or not a lot of content inside?
Reviewer 3 Report
The paper submitted to Micromachines reports an improved design for a micromotor-driven inchworm-like capsule robot which has enhancements in movement performance. Specifically, in order to maximize the periodic stroke and minimize the body length of their device, authors employed two axial compact expanding systems, a novel linear mechanism cooperating with the two expanding mechanisms, and a hollow cylinder-like power source, which takes full advantage of the free space around the linear mechanism, for wireless powering and transmission. Authors provided the overall design consideration comprehensively; the design parameters and working principle of each component in the system; and the experimental setups for testing the device performance in a transparent tube, and a suspended and tiled colon. Furthermore, the novelty of the device was also discussed carefully in the manuscript by comparing its movement performance and design parameters with other pre-existing devices. Overall the manuscript is very good, and I recommend this paper be accepted after addressing some minor revisions.
There are some comments for authors.
1) Some clumsy sentences and typos were found in the manuscript (i.e., lines 122-124, page 4). Please address this as well as other grammatical errors throughout the manuscript.
2) In the manuscript, authors mentioned that the power source would not cause any heating issue to the intestinal tissue by providing the change in the temperature for the power source operating continuously for 1 hour in a transparent tube. However, the heat absorbance rate of tissue is different from the plastic tube; therefore, in order to validate the safety of their device, it would be better to perform the temperature measurement of the power source operating for 1 hour in the colon instead.
3) The size of the ultimate device sounds large. Can the authors elaborate on the size of their device compared to pre-existing surgical equipment typically used during a colonoscopy?
Round 2
Reviewer 1 Report
Very good work. All remarks have been addressed and I strongly recommend it for publication
Reviewer 2 Report
Questions are answered well. NO further question!